# Fast and Simple Liquid Chromatography-Isotope Dilution Tandem Mass Spectrometry Method for Therapeutic Drug Monitoring of Dalbavancin in Long-Term Treatment of Subacute and/or Chronic Infections

**DOI:** 10.3390/pharmaceutics15020480

**Published:** 2023-02-01

**Authors:** Rossella Barone, Matteo Conti, Pier Giorgio Cojutti, Milo Gatti, Pierluigi Viale, Federico Pea

**Affiliations:** 1SSD Clinical Pharmacology Unit, IRCCS Azienda Ospedaliero Universitaria di Bologna, 40138 Bologna, Italy; 2Department of Medical and Surgical Sciences, Alma Mater Studiorum, University of Bologna, 40138 Bologna, Italy; 3Infectious Diseases Unit, IRCCS Azienda Ospedaliero Universitaria di Bologna, 40138 Bologna, Italy

**Keywords:** Dalbavancin, therapeutic drug monitoring, microsampling, Liquid Chromatography-Isotope Dilution Tandem Mass Spectrometry

## Abstract

Dalbavancin (DBV) is a long-acting antistaphylococcal lypoglycopeptide that is being increasingly used for long-term treatment of a wide range of subacute and/or chronic infections, mainly osteo-articular infections (OAI). Population pharmacokinetic studies showed that two 1500 mg doses 1 week apart can ensure effective treatment for several weeks. In this scenario, therapeutic drug monitoring (TDM) can be a helpful tool for providing clinicians with real-time feedback on the duration of optimal treatment by measuring drug concentrations over time in each single patient. The aim of this study was to develop and validate a fast and simple analytical method based on the Liquid Chromatography-Isotope Dilution Tandem Mass Spectrometry (ITD LC-MS/MS) technique for measuring DBV concentrations in human plasma microsamples. It will allow an innovative, very convenient and minimally invasive way of sampling. Analysis was performed by simple single-step sample preparation and very short instrumental run time (4 min). Analytical performance met all criteria in terms of specificity, sensitivity, linearity, precision, accuracy, matrix effect, extraction recovery, limit of quantification, dilution integrity and stability under different conditions set by the European Medicines Agency (EMA) for drug quantification by means of bioanalytical methods. The method was successfully applied for measuring DBV concentrations (range = 2.0–77.0 mg/L) in a cohort of patients receiving long-term DBV treatment of subacute and/or chronic infections.

## 1. Introduction

Dalbavancin (DBV) (Figure 1) is a new semi-synthetic antibiotic approved by both the Food and Drug Administration (FDA) and the European Medicines Agency (EMA) for the treatment of acute bacterial skin and soft tissue infections both in adults and in pediatrics [1]. Thanks to the very long elimination half-life, DBV is being increasingly used in clinical practice for the long-term treatment of a wide range of subacute and/or chronic infections, mainly osteo-articular infections (OAI). Population pharmacokinetic studies showed that two 1500 mg DBV doses 1 week apart could ensure effective treatment for several weeks [2,3]. However, it could be quite difficult establishing which could be the real duration of effective treatment due to the wide interindividual pharmacokinetic variability observed in this clinical scenario [2,3]. Therapeutic drug monitoring (TDM) can be a helpful tool for providing clinicians with real-time feedback on real duration of optimal treatment by assessing DBV concentrations over time in each single patient [2,3]. Nowadays, TDM is considered the only effective and safe way for optimizing exposure with antimicrobials, especially in critically ill patients [4,5]. By measuring concentrations in plasma, antibiotic dosages may be adjusted in real time for maximizing efficacy and minimizing the risk of resistance development [6]. Consequently, routine TDM of DBV could concur in improving the proper management of DBV use in long-term treatments [7,8]. The use of plasma microsampling techniques in TDM of long-term treatment regimens may have some practical advantages compared to conventional venipuncture methods [9]. Microsampling enables the reduction of stress and pain related to venipuncture procedures and may be particularly advisable in fragile populations [10,11]. Blood microsamples may be collected onto specialty paper(s) [11], polymeric tips or capillaries [9] and even directly collected onto dried plasma spots (DPS) [12]. Blood droplets can also be collected in small, dedicated vials, and plasma can be obtained by centrifugation in the laboratory.

In the field of pharmaceutical analysis, High Performance Liquid Chromatography (HPLC) is the most widely used analytical technique for antibiotic assays in biological samples and not only [13,14,15,16].

Currently, few are the methods that have been developed for TDM of DBV. One is based on high-performance liquid chromatography coupled with ultraviolet-visible detection (HPLC-UV-VIS) and has acceptable bias and precision but was tested only in three clinical cases for TDM-based dose optimization of DBV [17]. The other two methods were based on HPLC coupled with tandem mass spectrometry (LC-MS/MS) and used plasma sample aliquots of 50 or more microliters (µL) for the analysis [18,19,20].

The aim of this study was to develop and validate a fast and simple analytical method for measuring DBV concentrations in plasma microsamples by means of a liquid chromatography-isotope dilution tandem mass spectrometry (ITD LC-MS/MS) technique using as internal standard the exa-deuterated analog of DBV (DBV-d6), similar to Seraissol et al. [21]. 

## 2. Materials and Methods

### 2.1. Chemical and Reagents

DBV and DBV-d6 (internal standard—IS) powders were purchased from Sigma-Aldrich (Darmstadt, Germany) and Clearsynth Labs limited (Mumbai, India), respectively. The chemical structures of DBV and DBV-d6 are depicted in Figure 1. Methanol and formic acid (LC/MS grade) were purchased from CHROMASOLV™ (Thermofisher Scientific, Milan, Italy). LC–MS/MS grade water was produced by means of a Milli-Q^®^ Direct system (Millipore Merck—Darmstadt, Germany). Drug-free plasma from volunteers was supplied for control purposes by the IRCCS Azienda Ospedaliero Universitaria di Bologna (Bologna, Italy).

### 2.2. Stock Solutions, Standards and Quality Controls 

The DBV stock solution was prepared in MilliQ water at a concentration of 5 mg/mL. The working solution at 500 mg/L was obtained by water dilution of the stock solution and was used as the highest point of the calibration curve. All of the other calibration standards (0.125–5–25–50–250 mg/L) were prepared by subsequent dilution of the working solution with plasma. The final calibration range (0.125–500 mg/L) covered the range of plasma DBV concentrations expected on the basis of the clinical need. Quality control (QC) samples were set up from an independent stock of plasma. The QC samples were set at the following concentrations:-Low QC (LQC) = 7 mg/L;-Medium QC (MQC) = 70 mg/L;-High QC (HQC) = 280 mg/L.

IS solution of DBV-d6 was prepared at a concentration of 1 mg/L in methanol. IS and all the other working solutions were frozen at −80 °C.

### 2.3. Instrumentation 

LC-MS/MS analysis was performed by means of an Agilent 1295 UHPLC^®^ (Ultra High-Performance Liquid Chromatography) coupled with a quadrupole mass detector 6495c (Agilent, Santa Clara, CA, USA) equipped with an electrospray ion source (ESI). Chromatographic separation was achieved using a ZORBAX Eclipse plus C18 column (2.1mm width, 50 mm lenght, 1.8 µm particle size) (Agilent, Santa Clara, CA, USA). 

Mobile phase A, 0.1% formic acid in water, and mobile phase B, 0.1% formic acid in methanol were freshly prepared before the analysis. Gradient elution is described in Table 1, with mobile phase B’s initial proportion at 5%. Total run time was 4 min. Flow rate was set at 0.5 mL/min (see Table 1). The autosampler was cooled down to 10 °C, and the column temperature was set at 25 °C. The injection volume was 3 µL.

The MS was operated with positive ionization in Multiple Reaction Monitoring (MRM) mode, and the parameters are listed in Table 2. Optimized MS parameters were as follows: gas temp = 200 °C, gas flow = 14 L/min, nebulizer pressure = 35 psi, sheath gas temp = 300 °C, sheath gas flow = 11 L/min, capillary voltage = 4000 V, Nozzle voltage = 0 V. 

Chromatographic data acquisition, peak integration and quantification were performed by means of the MassHunter software version 10.0 (Agilent, Santa Clara, CA, USA).

### 2.4. Sample Pre-Treatment 

Plasma samples (3 μL) were added to 47 μL of ultrapure water to achieve a dilution factor of 17 and then mixed with 150 μL of the IS-methanol solution (1 mg/L DBV-d6). The mixture was vortexed for 15 s and then centrifuged at room temperature at 13,000 rpm for 5 min. Subsequently, 50 µL of the clear supernatant was transferred to an autosampler vial, and a volume of 3 µL was injected into the LC-MS/MS system (Figure 2).

### 2.5. Method Validation 

The analytical method was validated according to the EMA guidelines for validation of bioanalytical methods [10]. Selectivity, carry-over, linearity, accuracy and precision, lower limit of quantification (LLOQ), matrix effect, extraction efficiency and stability were evaluated.

#### 2.5.1. Selectivity 

A selective method should be able to quantify specifically the analyte in a complex mixture without any interference caused by endogenous and/or exogenous components present in the matrix. For testing selectivity, we analyzed 10 different plasma samples coming from different primary sources. 

#### 2.5.2. Carry-Over 

The carry-over effect was assessed by checking the eventual presence of peaks in chromatograms of blank plasma samples obtained after running the Upper Limit of Quantification calibrator (ULOQ, the highest calibrator concentration). Carry-over was considered as suitable whenever peaks were <20% of those recorded for the LLOQ samples.

#### 2.5.3. Linearity and Lower Limit of Quantification 

Calibration standards were prepared by spiking the blank matrix over the 0.125 to 500 mg/L range. Linearity of the calibration curve was confirmed by calculating the Percent Relative Error of back-calculated concentration (%RE). According to the fitness-for-purpose approach [22], to verify linear range, these relative errors must lie between ±15% for all the calibration range and between ±20% when limit of quantification (LOQ) is reached. The LLOQ was defined by the lowest calibrator in the selected dynamic range (0.125 μg/mL) and showed a signal-to-noise ratio (S/N) higher than 10.

#### 2.5.4. Precision and Accuracy 

Precision (mean CV%) and accuracy (mean BIAS%) were assessed by extracting and analyzing four concentration levels (LLOQ, LQC, MQC and HQC) for five times both in the same day (intra-day) and in three different days (inter-day).

#### 2.5.5. Matrix Effect and Extraction Recovery 

Percent Matrix effect (ME) and Extraction Recovery (ER) were calculated at Low, Medium and High concentration levels by means of the following equations:ME (%) = B/A × 100;
ER (%) = C/B × 100
where:

A = DBV over DBV-d6 mean peak area ratio obtained by injecting water samples (N = 3) spiked at the three concentration levels. 

B = DBV over DBV-d6 mean peak area ratio drug-free matrix extracts (N = 3) spiked at the three concentration levels **after** extraction. 

C = DBV over DBV-d6 mean peak area ratio drug-free matrix extracts (N = 3) spiked at the three concentration levels **before** extraction.

Both ME and ER were tested in 10 different patients’ plasma samples for addressing individual matrix composition variability.

#### 2.5.6. Stability 

Stability of DBV was assessed by comparing the nominal concentrations of LQC, MQC and HQC with those observed under two different conditions: sample extracts kept on board at 10 °C for 24 h and frozen at −80 °C for 24 h;plasma samples analyzed after three complete freeze and thaw cycles (from −80 °C to 25 °C).

Stability in the above operating conditions was deemed suitable if DBV concentrations were within ±15% of the nominal value. 

### 2.6. Clinical Application

The presented ITD LC-MS/MS method was tested for measuring DBV concentrations in patients receiving DBV for the treatment of long-term subacute or chronic documented Gram-positive infections (namely osteoarticular infections, endocarditis, and endovascular prosthetic infections). In our study, DBV treatment was started according to the internal protocol consisting of two 1500 mg doses one week apart. Requirement for additional doses was assessed by the treating physician after at least four weeks according to the underlying clinical conditions of each patient. The study was conducted according to the guidelines of the Declaration of Helsinki and approved by the local ethical committee [No. 897/2021/Oss/AOUBo on 29 November 2021]. Informed written consent was waived due to the retrospective and observational nature of the study [2,3]. The number of samples assessed was 520. Samples for Dalbavancin TDM were usually collected at predefined timeframes between 21 and 35 days after first administration. More details on this may be found in our previous study [3]. Blood samples respected the cold chain delivery and were processed either immediately after delivery or after freezing at −80 °C until analysis, depending on case-by-case. 

## 3. Results 

### 3.1. Optimization of LC-MS/MS Conditions

LC-MS/MS conditions were set for granting good DBV peak shape and quality regardless of the very short chromatographic run time. For this purpose, we selected the ZORBAX Eclipse plus C18 column (2.1mm width, 50 mm lenght, 1.8 µm particle size) (Agilent, Santa Clara, CA, USA), as suggested previously [16].

A mobile phase consisting of (A) water-formic acid (100:0.1, *v/v*) and (B) methanol-formic acid (100:0.1, *v/v*) was applied with elution gradient, as described in Table 1. ESI parameters were optimized by using the Optimizer software version 10.0 (Agilent), monitoring the [M + 2H] + 2 precursor ion intensity signals. Three microliters of DBV or DBV-d6 solution at 1 µg/mL concentration were injected into the instrument while a mobile phase A-B 50-50% was flowing at 0.1 mL/min. For optimizing sensitivity, mass transitions of doubly charged ions were selected at *m*/*z* 909.2 → 340.1 for DBV and at *m*/*z* 912.3 → 340.2 for DBV-d6, from the MS/MS fragmentation spectra experimentally obtained (Figure 3) and were in accordance with those obtained in a previous study [18]. 

Drug-free plasma sample MRM chromatograms (Figure 4a) extracted with IS-methanol solution showed only the presence of the DBV-d6 peak without the DBV peak, thus confirming the purity of the DBV-d6 standard solution that we used as IS and that no isobaric interferences occurred for DBV.

Total chromatographic run time was as short as 4 min, but this did not compromise the chromatographic performance. The retention time (rt) was 2.43 min and was very reproducible throughout analytical runs, thus confirming that column reconditioning after the gradient runs was optimal. Consequently, a reconditioning step of 0,5 min in the gradient (see Table 1) was thought to be enough for proper column reconditioning between runs.

LLOQ sample MRM chromatograms (Figure 4b) showed a signal-to-noise (S/N) ratio of 144.3 for the DBV peak. This shows the high sensitivity of the method, with the S/N value being even higher than that of the LOQ used for current validation (0.125 mg/L).

Real sample MRM chromatograms (Figure 4c) showed that peak shape and resolution were optimal even at very low concentrations. Isobaric peaks were never observed in samples, and this confirms the selectivity of the selected MRM transitions.

### 3.2. Method Validation 

#### 3.2.1. Selectivity 

Ten DBV-free plasma samples coming from different sources were scrutinized for checking the eventual presence of interfering peaks in the MRM chromatograms obtained by running the chromatographic gradient. All MRMs of drug-free plasma samples (example in Figure 4a) showed only the DBV-d6 peak without any DBV peak. This ruled out the possibility that DBV could have been a potential contaminant of the DBV-d6 solutions and confirmed the good quality of the DBV-d6 IS.

#### 3.2.2. Carry-Over 

MRM chromatograms of drug-free plasma samples injected after running the high calibration level (HCL = 500 µg/L) showed no peak for DBV, thus confirming that carry-over was negligible. This could probably be favored by the step at 95% of mobile phase B (see Table 1) in the gradient employed for elution.

#### 3.2.3. Lower Limit of Quantification and Linearity 

The LLOQ was 0.125 μg/mL and corresponded to the lowest point of the calibration curve. Its S/N value was 95.0 and was much higher than the usually considered LOQ of 10.

The calibration curve model using response (= DBV peak area/DBV-d6 peak area) over concentration showed good data fitting (example in Figure 5). The equation calculated by pooling data obtained in seven different days was y = 0.0432x − 0.0137. The average regression coefficient was R^2^ = 0.9997. Percent relative error of back-calculated concentrations (%RE) ranged from 18% at LLOQ to an average of 8,7% for all the other calibrators. 

#### 3.2.4. Precision and Accuracy 

Precision (mean CV%) and accuracy (mean BIAS%) results are shown in Table 3. The intra- and inter-day coefficients of variation of the different QC levels ranged from 7% to 11% and from 7% to 13%, respectively. Likewise, the intra- and inter-day accuracy bias of the LQC, the MQC and the HQC ranged from 9% to 10% and from 7.9% to 14%, respectively. 

#### 3.2.5. Matrix Effect and Extraction Recovery 

Percent Matrix effect (%ME) and percent Extraction Recovery yield (%ER) were calculated at Low, Medium and High concentration levels (Table 4). A signal enhancement effect was observed at the low and medium concentrations, whereas a slight signal suppression effect was observed at the high concentrations. This finding, coupled with that of an extraction recovery yield increase which was proportional to DBV concentrations (from 83.3% to 97.7%), pointed out that the addition of the IS is really needed for providing reliable quantification of DBV concentrations throughout the whole dynamic investigated range.

#### 3.2.6. Stability 

DBV stability was tested for all of the QC levels in different operating conditions as specified in Table 5. After two freeze and thaw cycles, DBV concentrations decreased at all of the levels tested. This pointed out that sample reprocessing is unfeasible. Autosampler extracts kept at 10 °C were stable for at least 24 h, and this may allow useful sample injections when needed. Autosampler extracts were stable after frozen at −20 °C for more than 48 h, and this may allow the opportunity to prepare samples on different days which is a useful option for laboratory working management.

### 3.3. Clinical Application 

The DBV concentrations measured by applying the ITD LC-MS/MS method in samples collected from patients receiving DVB for long-term treatment of subacute or chronic documented Gram-positive infections were all within the calibration range. The widespread distribution of DBV plasma concentrations (see Figure 6 below) that were observed in more than 500 TDM assessments may support the usefulness of this approach for individualizing DBV exposure and attaining optimal PK/PD targets in each single patient.

## 4. Discussion

In this study, we developed and validated an analytical method for fast, accurate and precise quantitative determination of DBV in human plasma microsamples of 3 μL, by means of an ITD LC-MS/MS technique. Such small microsamples could be obtained by finger or heel pricking and may favor TDM application in long-term treatment regimens. It could be performed during day-hospital accesses or even at home and may minimize the need for venipuncture procedures, the patients’ discomfort and the sample logistic issues [9,10,11]. 

Thanks to the high analytical performance, the proposed method met all of the criteria in the EMA guidelines [23], with the exception of QC Low extract analyzed 24 h after 10 °C-autosampler storage, which shows an Average Accuracy Bias (%) slightly above the 15% (22.1%) required from the EMA guidelines. It was successfully validated and applied for TDM of DBV concentrations in clinical practice. The high specificity of this method is probably related to the MRM transitions that we selected, which are the same used by Alebic-Kolbah et al. but different from those employed by others [19,20]. The fact of adopting a different combination of extraction solvents in our methods could have potentially brought different interfering endogenous or exogenous compounds in the analysis. Indeed, this was not the case and high specificity was verified and confirmed. It must be recognized that the selected MRM transitions are specific for the B_0_ and B_1_ forms of DBV and that the homologous A_0_, A_1_ and B_2_ DBV forms are not taken into account. However, this is not an issue as these forms are not prevalent in the pharmaceutic formulations currently used in humans.

The method also showed high sensitivity, even potentially much higher than the LLOQ set for validation purposes (at 0.125 mg/L), which is well below the DBV trough levels expected for granting efficacy (namely 4.02 or 8.04 mg/L) [3]. The LLOQ of our method is similar to that of the method of Alebic-Kolbah et al. (0.1 mg/L), but the sample volume used by them (50 µL) was much higher than that used by us (3 µL). Thus, the present method is much more sensitive than those published so far [17,18,19,24]. 

The calibration range of our method is extended and dynamic (0.125–500 mg/L) and may cover the entire range of concentrations expected in clinical practice. This makes feasible always measuring DBV concentrations in patients’ samples directly, regardless of them being peak or through values, with no need for sample dilution or re-run even in the presence of very high concentrations.

DBV has very high plasma protein binding, so that its extraction from human plasma could be challenging [25]. Several solvent extraction-precipitation procedures can be adopted for DBV recovery from biological samples, such as ACN:MeOH (50:50) [19], ACN:H_2_O (20:80) [18], or pure methanol [20]. We used for extractive protein-crash down the MeOH:H_2_O (3:1 *v*/*v*) mixture in a stepwise fashion after diluting the microsamples with water (1:17 *v*/*v*), as this procedure may facilitate the handling of very small sample volume like ours. The extraction recovery of our method was high (about 87%) at all of the concentration levels (Table 4), and resulted even higher than those observed previously in other studies in which different extraction solvent combinations were used [18,19,20]. 

We used a deuterated chemical analog of DBV as IS. This is in agreement with the study of Seraissol et al. [21] and different from other methods published previously [17,18,19]. The major advantage granted by using a deuterated IS is the high accuracy achieved in compensating for extraction yield and matrix effect variations of patients’ samples whose content and composition of endogenous and exogenous compounds may be very different.

It is noteworthy that the good stability of DBV observed both in plasma samples and in extracts under our operative conditions is an important feature. Stability in plasma samples allows reliable results even in samples coming from remote clinical centers, whereas stability in extracts allows re-analysis by simple re-injections.

Finally, another interesting aspect of this method is that it is based on microsamples so that it could be easily implemented for TDM purposes also by using next-generation microsampling techniques, such as Volumetric Absorptive Microsampling (VAMS) [26] DPS [27] and/or other microfluidic devices [9].

## 5. Conclusions

In conclusion, we developed a fast, sensitive, accurate, precise and reliable ITD LC-MS/MS method for measuring DBV in human plasma microsamples. The method may be successfully applied for routine TDM of DBV in patients undergoing long-term treatment for subacute and/or chronic staphylococcal infections.

## Figures and Tables

**Figure 1 pharmaceutics-15-00480-f001:**
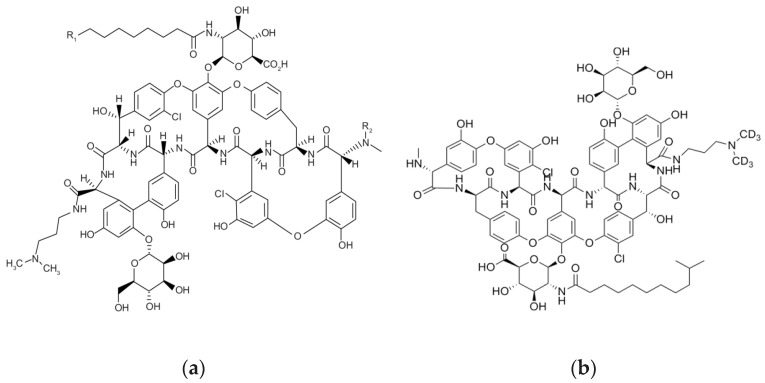
Chemical structure of Dalbavancin (**a**) and of the exa-deuterated analog Dalbavancin-d6 (**b**) used as Internal Standard (IS).

**Figure 2 pharmaceutics-15-00480-f002:**
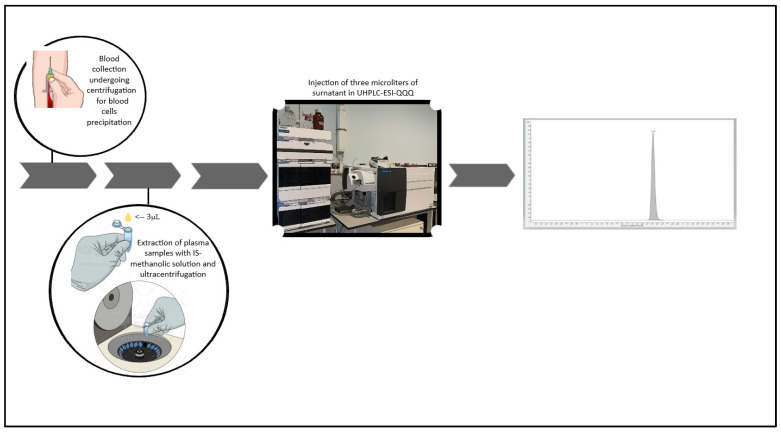
Images above describes the working flow of the developed method.

**Figure 3 pharmaceutics-15-00480-f003:**
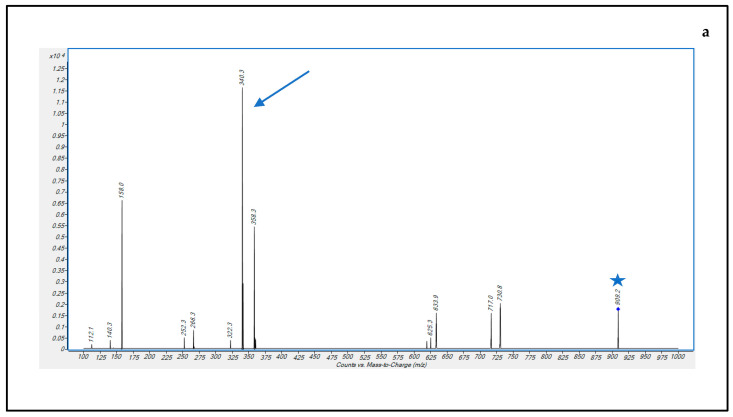
MS/MS fragmentation pattern spectra of DBV (**a**) and DBV-d6 (**b**) Precursor [M + 2H] + 2 ions and product ions selected for MRM transitions are indicated by a blue star and a blue arrow, respectively.

**Figure 4 pharmaceutics-15-00480-f004:**
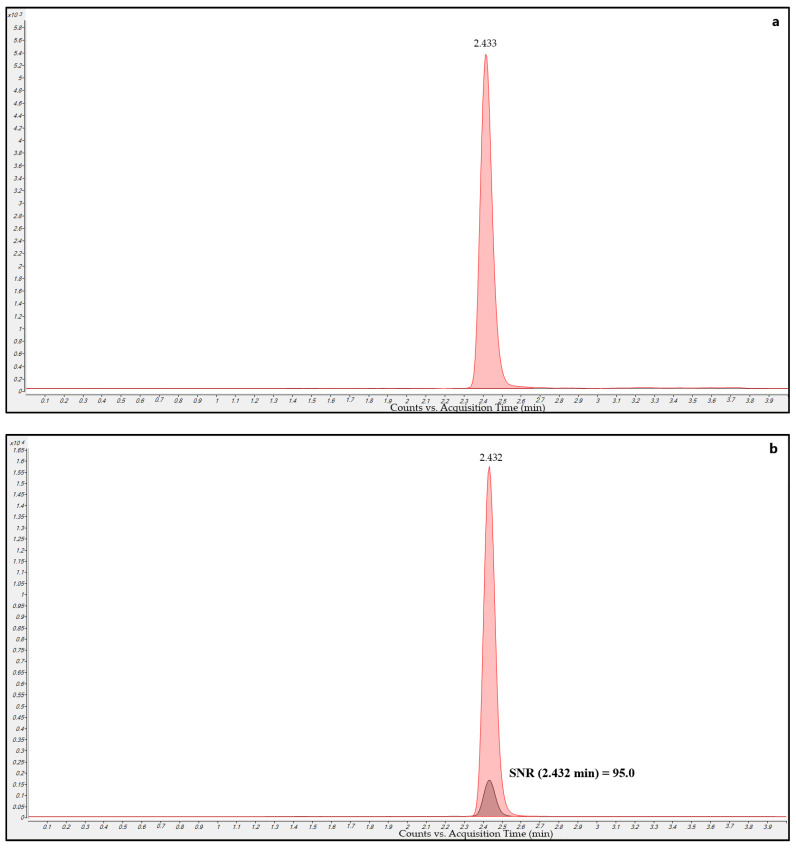
Overlayed MRM chromatograms for DBV (black) and DBV-d6 (red) obtained in the analysis of different samples. (**a**) blank sample extracted with the methanol-IS solution showing a well-defined DBV-d6 peak without any DBV peak; (**b**) LLOQ sample with the respective S/N ratio (SNR); (**c**) patient sample collected 25 days after starting DBV showing good peak shape and resolution for both DBV and DBV-d6.

**Figure 5 pharmaceutics-15-00480-f005:**
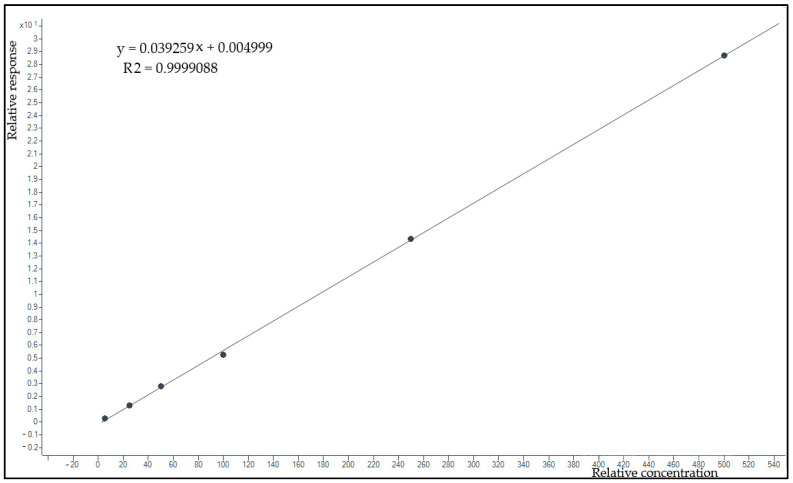
Example of a calibration curve obtained by plotting the DBV/DBV-d6 area ratio (response) over concentration in the 0.125–500 mg/L range. The linear fitting of the six calibration points and the very high correlation coefficient show the stringent linearity of the calibration model.

**Figure 6 pharmaceutics-15-00480-f006:**
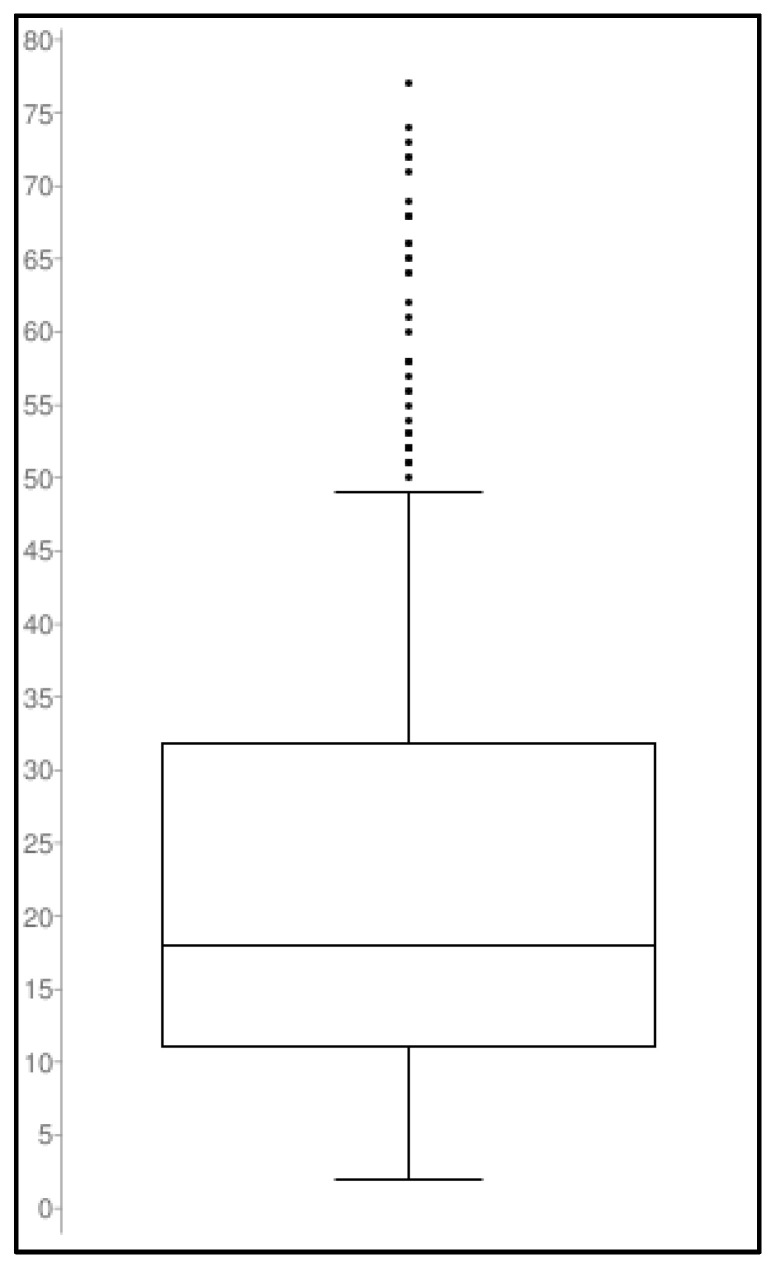
Box plot showing the spread of the DBV concentration measured in 520 patients’ plasma microsamples. Population size: 520; Median: 18; Minimum: 2.0; Maximum: 77.0; First quartile: 11; Third quartile: 31.8; Interquartile Range: 20.8.

**Table 1 pharmaceutics-15-00480-t001:** Binary pump program used for fast constant flow elution with mobile phases A and B, as described in methods.

Time (min)	A (%)	B (%)	Flow (mL/min)
0.00	95	5	0.500
2.00	30	70	0.500
2.50	5	95	0.500
3.00	5	95	0.500
3.01	95	5	0.500
4.00	95	5	0.500

**Table 2 pharmaceutics-15-00480-t002:** Specific MRM transition parameters used for DBV and for DBV-d6 acquisition (RT = retention time).

Analyte	R T (min)	Precursor Ion (*m*/*z*)	Product Ion (*m*/*z*)	Dwell Time (ms)	Fragmentator (V)	Collision Energy (V)
DBV	2.40	909.3	340.2	200	166	28
DBV-d6	2.42	912.3	340.2	200	166	28

**Table 3 pharmaceutics-15-00480-t003:** Intra-day and inter-day average (avg) precision and accuracy assessed at four concentration levels (LLOQ, LQC, MQC and HQC) for five times (intra-day) in three different analytical runs (inter-day).

		Intra-Day (*n* = 5)			Inter-Day (*n* = 3)		
QC Levels	Nominal Conc. (μg/mL)	Avg Conc. (μg/mL)	Avg Precision (CV%)	Avg Accuracy (Bias%)	Avg Conc. (μg/mL)	Avg Precision (CV%)	Avg Accuracy (Bias%)
LLOQ	0.125	0.130	12.3	4.1	0.136	13.5	7.2
Low	7	6.55	10.2	9.1	6.41	7.6	9.3
Medium	70	68.3	8.4	5.8	64.9	7.0	7.9
High	280	252.4	11.6	9.9	241.7	13.0	13.7

**Table 4 pharmaceutics-15-00480-t004:** Average (Avg) Matrix effect (ME%) and Recovery (ER%) of DBV measured at different concentration levels.

Quality Control Level	N°	Avg Me (%)	Avg IS-Normalized Me (%)	Avg ER (%)
LQC	30	109.9	114.9	86.3
MQC	30	105.7	115.5	83.5
HQC	30	87.6	93.6	97.7

N° (number of plasma samples tested: 10 different patients’ plasma samples per each level of concentration).

**Table 5 pharmaceutics-15-00480-t005:** Stability of DBV at different storage conditions. In our study, we tested both the extracts and the plasma samples (according to routine needs).

Quality Control		Low	Medium	High
Types of Sample	Tested Conditions	Avg Accuracy (Bias%)	Avg Accuracy (Bias%)	Avg Accuracy (Bias%)
extract	autosampler post 24 h	22.1	9.4	14.8
	freezer post 24 h	16.8	6.7	12.3
plasma samples	**freeze-thaw stability**			
	1 cycle	14.8	8.2	11.4
	2 cycle	15.1	9.3	13.6
	3 cycle	17.4	15.7	16.4

## Data Availability

The data presented in this study are available on request from the corresponding author. The data are not publicly available due to privacy concerns.

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
