# Peer review of "Fast and Simple Liquid Chromatography-Isotope Dilution Tandem Mass Spectrometry Method for Therapeutic Drug Monitoring of Dalbavancin in Long-Term Treatment of Subacute and/or Chronic Infections"

_pharmaceutics, 2023, doi:10.3390/pharmaceutics15020480_

Round 1

Reviewer 1 Report

very nice study. Well chosen methods, results presented clearly, well discussed. The objectives of the work have been achieved - the method of determination has been presented.
I actually have one thing to say about work. I think the introduction is too general. The DBV description itself is good and appropriate, but limiting the definition of TDM to one sentence is too short. Thus, the reference to the use of TDM of DBV in the applications becomes an irrelevant cliche. Please define TDM, add one sentence about the goals (mainly to improve the safety of therapy, in selected patients also effectiveness), refer to the classics of TDM, e.g. digoxin and carbamazepine - 20-year observational studies are available) or to new ones like direct oral anticoagulants (8-years observational studies are available). I would single out this passage as a subsection of the introduction. Such an extension is necessary because even in the title there is TDM ... and the publication will also be read by clinicians.
I expect that after the introduction of such an extension, the work will be considered for publication.

Author Response

Response to reviewer

Thanks for your comments and suggestions. We insert some sentences regarding TDM in the introduction (page 2).

Reviewer 2 Report

The authors describe a novel method for the quantification of dalbavancin using a very small sample of plasma.

Some minor points need to be addressed:

In the abstract they mention "a very innovative sampling method"but it is not specified in the text. Also from the graphical abstract it seems thay are using capillary blood but it's not specified in the text

The authors say that the method has been applied on several clinicl samples but no information is provided in the text

Paragraph 2.3 and 2.4 are too schematic: please specify better

Author Response

Response to reviewer

Thanks for your comments and suggestion, our responses point-to-point are listed below:

  1. We added in the introduction a paragraph (pag.2) to explain the actual experimental settings and accordingly changed the graphical abstract.
  2. In the paragraphs 2.6 e 3.3 we gave information regarding the application of the validated method on 520 samples. Moreover, we elaborate a boxplot representing the distribution of patient’ plasmatic concentration.
  3. Paragraph 2.3 and 2.4 have been modified as recommended.

Reviewer 3 Report

The work is average but may be improved by the inclusion of the following suggestions.

1.      English should be improved throughout the manuscript.

2.      Quantitative information should be provided in the abstract.

3.      The concussion should be concise and to the point indicating the application of the work.

4.      The novelty of the work should be established.

5.      Please write one paragraph in the introduction about HPLC importance, in general, and you can consult the following articles to make this manuscript more useful to the readers.

J. Pham Biomed Anal 48, 175-188 (2001).; J. Sepn. Sci., 37, 1033-1057 (2014); J. Sepn. Sci., 35, 3235-3249 (2012); Fres. J. anal. Chem., 370, 951-955 (2001).

8.  Please provide error graphs in the figure; where are required.

9.      Please improve the quality of the Figures.

10.  Please compare your results with previous studies and mention clearly how your work is important in comparison to already been reported.

Author Response

Response to reviewer

Thanks for your comments and suggestion, our responses point-to-point are listed below:

  1. We have subjected the manuscript to careful English revision.
  2. Quantitative information are now provided in the abstract.
  3. Thank you for this suggestion. The application of the work consisted in the validation of an accurate and reliable quantification method starting from a very small amount of plasma.
  4. Thank you for this relevant comment, allowing us to better clarify the novelty of our work. The novelty of the work have been established in the sensibility of the method and in the very small amount of plasma used for the analysis that may favor TDM application in long-term treatment regimens. Opening the possibility of TDM through microsampling devices, thanks to the ability of quantifying DBV in very small biological microsamples.
  5. We added some sentences regarding the importance of HPLC and mass spectrometry.
  6. We didn’t present graphs requiring error bars. In the calibration curve (Fig. ) we provided an example of calibration graph instead of all the calibration curves and the relative error bars.
  7. The graphical resolution of the figures has been improved.
  8. Thank you for this relevant comment. As suggested, we compared our results with previous studies (Avataneo V. et al; Zhu D. et al; Alt S. et al; Chiriac U. et al; Alebic-Kolbah T.; Seraissol et al.) in the discussion.

Reviewer 4 Report

The manuscript (article) presents a LC-MS/MS method for analysis of dalbavancin. The authors stated that the method can be used in therapeutic drug monitoring of this lipoglycopeptide antibiotic in long-term treatment of infections. The drug was approved by the FDA in 2014 and represents a new tool for methicillin-resistant Staphylococcus aureus treatment. The topic is interesting and fits with the special issue of the journal (Pharmaceutics), but the manuscript fails in the method development, validation and presentation of the information. The manuscript has serious flaws.

Introduction:

The analytical methods developed by Seraissol et al. (J Pharm Biomed Anal. 219 (2022) 114900) and Deng et al. (J Chromatogr A 1538 (2018) 54-59) should be mentioned and discussed. The method developed by Seraissol et al. uses isotopically labelled internal standard of dalbavancin. Analytical parameters of the previously published methods should be discussed.

Figure 2 should be used as Graphical abstract.

2.3 Instrumentation

Injection volume 3 uL – it should be 3 µL

Column – temperature kept ai 25…. – it should be …. kept at 25….

3 Inlet

Water-formic acid (100:0.1, v/v) – I recommend to use 0.1% formic acid in water

Methanol-formic acid (100:0.1, v/v) – I recommend to use 0.1% formic acid in methanol

4 MS/MS

Gas temperature 200 C – it should be 200ºC

The optimization of the method is missing – there is no information how was the column selected, optimization of the mobile phase selection is missing, optimization of the detection step is also not presented

The MS spectra of DBV and DBV-d6 are missing – selection of the parent and product ions is not presented

How was performed the optimization/selection of the ESI and MS parameters (e.g. nebulizing pressure, gas temperature, gas flow, capillary voltage, etc.)?

2.6 Clinical application

….patients receiving DLB…. – it should be ….patients receiving DVB….

Information about the application of the drug is missing. Information about the dosage form is missing. Number of patients is not presented.

Sampling procedure is not described. How was the sample collected. Preanalytical and sample treatment steps are not mentioned.

Results

Please, do not use comma in the number – e.g. 0,5 – point should be used – i.e. 0.5

How was confirmed the linearity of the method?

Was the LLOQ value tested experimentally?

3.2.4 Precision and accuracy

Tables – do not use comma in the number

Table 4 – explain the abbreviation No

Table 5 – avg accuracy (bias%) for LQC was 22.1% - is this value acceptable? Is it within the EMA or FDA guideline demands?

The results of real application are missing. Number of samples, determined concentration in the samples are not mentioned.

Ethics Committee statement is missing.

Author Response

Response to reviewer

Thanks for your comments and suggestion, our responses point-to-point are listed below:

  • Thank you for this relevant comment. We mentioned in the introduction section the suggested articles;
  • Figure 2 has been used as graphical abstract;
  • Paragraph “2.3 Instrumentation”has been corrected according to your relevant suggestions;
  • We thank the reviewer for this important comment, allowing us to better clarify these relevant methodological issues. Paragraph “1 Optimization of LC-MS/MS conditions” was added in order to provide additional details concerning the optimization of the method.
  • Thank you for this comment. We added a specific section (Paragraph “6 Clinical application”) provided details on clinical use of dalbavancin.
  • Results: All the errors have been corrected. The linearity was confirmed by 6-point calibration curve always showing R2 >0.99. The LLOQ was tested evaluating the signal-to-noise ratio > 10.
  • “3.4 Precision and accuracy”. All the errors have been corrected. In the discussion the Average Accuracy Bias (%) slightly above the 15% (22.1%) has been discussed (page 18).
  • The results have been added in the paragraph 3.3 Clinical application.
  • Ethic Committee statement is now present in “2.6 Clinical application”.

Round 2

Reviewer 1 Report

may be considered for publication in its current form

Author Response

Thank you for appreciating the revised version.

Reviewer 3 Report

Accepted

Author Response

Thank you for appreciating the revised version of the manuscript.

Reviewer 4 Report

Page 3 - Materials and methods: ...system (Millipore Merck - Darmstadt, Germania) ... it should be Germany!!!

Page 4: please explain the abbreviations LQC, MQC, HQC

Page 11: please explain the abbreviation HCL

Page 11: 3.2.3 Lower limit of quantification and Linearity: you mentioned that the S/N ratio at the LLOQ concentration level was 144.3, but in the Figure 4b, the S/N ratio at the LLOQ level is 95 - explain this discrepancy!

Page 11: 3.2.3 Lower limit of quantification and Linearity: The parameter R2 do not confirm linearity of the developed method. Confirm the linearity of the method by an statistic method.

Author Response

We thank the reviewer for appreciating the revised version of the manuscript. The manuscript has been corrected according to reviewer's suggestions as follows:

Point 1: Page 3 - Materials and methods: the error has been corrected.

Point 2: Page 4: the abbreviations LQC, MQC, HQC has been explained.

Point 3: Page 11 the abbreviation HCL has been explained

Point 4: Page 11: The value 144.3 was a typo. The correct value is 95 and has been corrected in the revised version of the manuscript.

Point 5: Page 11: 3.2.3 Lower limit of quantification and Linearity: The linearity of the calibration curve has been statistically tested by fitness-for-purpose approach and not by considering the R2

Round 3

Reviewer 4 Report

Thank you for corrections and for answering all my questions.